# Role of Cannabidiol for Improvement of the Quality of Life in Cancer Patients: Potential and Challenges

**DOI:** 10.3390/ijms232112956

**Published:** 2022-10-26

**Authors:** Ryan Green, Roukiah Khalil, Shyam S. Mohapatra, Subhra Mohapatra

**Affiliations:** 1Department of Molecular Medicine, Morsani College of Medicine, University of South Florida, Tampa, FL 33612, USA; 2Department of Internal Medicine, Morsani College of Medicine, University of South Florida, Tampa, FL 33612, USA; 3James A Haley VA Hospital, Tampa, FL 33612, USA

**Keywords:** cannabidiol, cancer, pain, clinical trials

## Abstract

There is currently a growing interest in the use of cannabidiol (CBD) to alleviate the symptoms caused by cancer, including pain, sleep disruption, and anxiety. CBD is often self-administered as an over-the-counter supplement, and patients have reported benefits from its use. However, despite the progress made, the mechanisms underlying CBD’s anti-cancer activity remain divergent and unclear. Herein, we provide a comprehensive review of molecular mechanisms to determine convergent anti-cancer actions of CBD from pre-clinical and clinical studies. In vitro studies have begun to elucidate the molecular targets of CBD and provide evidence of CBD’s anti-tumor properties in cell and mouse models of cancer. Furthermore, several clinical trials have been completed testing CBD’s efficacy in treating cancer-related pain. However, most use a mixture of CBD and the psychoactive, tetrahydrocannabinol (THC), and/or use variable dosing that is not consistent between individual patients. Despite these limitations, significant reductions in pain and opioid use have been reported in cancer patients using CBD or CBD+THC. Additionally, significant improvements in quality-of-life measures and patients’ overall satisfaction with their treatment have been reported. Thus, there is growing evidence suggesting that CBD might be useful to improve the overall quality of life of cancer patients by both alleviating cancer symptoms and by synergizing with cancer therapies to improve their efficacy. However, many questions remain unanswered regarding the use of CBD in cancer treatment, including the optimal dose, effective combinations with other drugs, and which biomarkers/clinical presentation of symptoms may guide its use.

## 1. Introduction

Cannabidiol (CBD) is a phyto-alkaloid isolated from plants in the Cannabaceae family and genus Cannabis. Depending on the species and extraction method, CBD can comprise as much as 90% of the total plant extract, which also contains many additional terpenes and cannabinoids, including tetrahydrocannabinol (THC) [1]. Some medical preparations of the plant extract contain both THC and CBD; however, the psychoactive properties of THC are known to cause intoxication and impairment, which limit the use of the drug [2,3]. CBD is not psychoactive and, thus, suffers no such limitation. Recently, many pre-clinical and clinical studies have been undertaken to determine the effects of CBD on pain, sleep, appetite, anxiety, and cognition, and its potential to treat movement disorders, including seizures [4,5,6,7,8]. This review focuses on the current interest in the use of CBD as a palliative or combination therapy for cancer, as its pleiotropic physiological effects may serve to counteract symptoms of cancer or side effects of chemotherapies [9,10].

To increase the understanding of CBD action, there is a growing body of in vitro and pre-clinical evidence for the effectiveness of CBD in slowing tumor growth and causing cancer cell death [11,12,13,14,15]. CBD is metabolized by the cytochrome P450 (CYP) family of enzymes in the liver, with CYP2C9, CYP2C19, and CYP3A4 responsible for most of its conversion to 7-OH-CBD [16,17].

CBD is a generally well tolerated drug and, thus, the dose range can be wide, with different studies or indications prescribing 20–1500 mg per day [6,9,10,18,19,20]. CBD has the potential to slow the metabolism of other drug substrates of the CYP enzymes, and the pharmacokinetics of any such combinations should be closely monitored. Despite promising case studies, the anti-cancer effects of CBD have not been explored in patients using systematically designed randomized controlled trials [21,22]. Moreover, currently available clinical results of CBD’s use in cancer patients either have a small sample size, are not well (placebo) controlled, or employ various CBD doses, formulations, or drug combinations.

Thus, there is currently an unmet need for additional studies to determine the optimal dose of CBD to effectively treat specific symptoms, the safety of its use in combination with specific cancer therapies, and the extent of its anti-tumor properties. To address this need, herein, we have comprehensively reviewed and critically appraised the evidence, and provided an unbiased view of the potential and current challenges.

## 2. Methodology for Search and Evaluation of Literature

In order to evaluate the current state-of-the-art regarding the use of CBD as a treatment for cancer in vitro, in vivo, or in the clinic, we performed multiple searches of scientific literature databases, including PubMed.gov (NIH), Google Scholar, ClinicalTrials.gov, and the EU Clinical Trials Register. Searches were performed between 1 August 2022 and 1 October 2022. Search terms included cannabidiol, cancer, CBD, tetrahydrocannabinol, THC, mice, in vivo, clinical trial, pain, and palliation.

A search of PubMed.gov for the keywords “cannabidiol” and “cancer” yielded 418 records published in the years 2018–2022, including 140 reviews. A search of Google Scholar for the same terms and time period yielded 13,500 records, including 3820 classified as review articles. Of the PubMed results, 81 included the keyword “vitro”, and 67 included the keyword “cell line”, which indicated they were likely to be in vitro studies. Similarly, 2800 of the Google Scholar results included the keyword “cell line”, and 7510 results included the keyword “vitro”. An initial review of the titles and abstracts yielded several common mechanistic themes which were used as added keywords (AND operator) for subsequent searches on Google Scholar and PubMed, respectively, with the following numbers of results: “cell cycle arrest”, 942 (9); “reactive oxygen species”, 2880 (19); “pyroptosis”, 158 (2); and apoptosis, 4840 (73). Examples of studies describing these mechanisms of CBD activity in cancer are discussed below. A search of ClinicalTrials.gov for the condition or disease, “cancer”, and the other terms, “cannabidiol”, and screening for trials that have posted results yielded 10 studies. Three of the results were excluded from analysis, since they aimed to treat seizures in patients diagnosed with Sturge–Weber syndrome or Tuberous Sclerosis. The details of the remaining seven clinical studies of CBD in cancer patients are analyzed and presented herein. A search of the EU Clinical Trials Register for the keywords “cannabidiol” and “cancer” and screening for only trials which have posted results yielded five trials corresponding to the European arms of the multicenter Sativex ^®^ studies.

## 3. Biology and Molecular Targets of CBD

The multitude of physiological effects precipitated by CBD can be attributed to its many cellular molecular targets, with over 75 cell-surface and intracellular protein interactions described thus far [4]. The most well-characterized of these is a pair of G protein coupled receptors (GPCRs) termed the “cannabinoid receptors” 1 and 2 (CB-1/2), encoded by the genes CNR1/2 [23]. These receptors initiate downstream signaling through phospholipase C, adenylate cyclase, and beta arrestin to activate protein kinase C, cAMP-dependent protein kinase, and ERK1/2, which control cellular processes such as apoptosis, cell cycle progression, and autophagy [24]. CB-1 is known to inhibit Wnt/β-catenin signaling, which is a driver of epithelial-to-mesenchymal transition in cancer and the cancer stem cell phenotype [25]. CB-2 may have a role in the regulation of angiogenesis in tumors via CXCR4 and ICAM-1 signaling [26,27]. CBD is also known to bind to additional G-protein coupled receptors, including GPR 3, 6, 12, 55, µ/δ-opioid receptors, and the serotoninergic 5HT1a receptor (5HT1a) [28].

CBD also targets cell surface transporters and ion channels, including the drug transporters ABCC1 and ABCG2, the inhibition of which may contribute to the synergy between CBD and anti-cancer drugs [29,30]. CBD has been shown to inhibit the nucleoside transporter ENT, the Mg ATPase, and multiple fatty acid binding proteins, including FABP1, 3, 5, and 7 [31,32]. CBD can activate transient receptor potential channels in the vanilloid family (TRPV) 1–4 at nanomolar to micromolar concentrations [33]. These channels are present on a wide range of cell types and can activate downstream calcium signaling, promoting apoptosis. CBD is also an allosteric regulator of ligand gated ion channels in the GABAa family and an inhibitor of calcium/sodium gated ion channels in the micromolar to the nanomolar range [34,35]. The distribution of CBD receptor expression within the body is depicted in Figure 1 [36].

The intracellular targets of CBD (Figure 2) include components of the mitochondrial electron transport chain complex 1, 2, and 4; several cytochrome P450 enzymes; the pro-inflammatory enzymes COX1, COX2, and LOX5; indolamine 2,3 dioxygenase; and peroxisome proliferator-activated receptor gamma (PPAR-γ) [16,37,38,39,40,41]. However, only the CYP450 enzymes are inhibited at nanomolar concentrations, whereas the others have EC50 values in the micromolar range, which has not been demonstrated in human plasma. CBD is known to have general anti-inflammatory effects, some of which may be mediated by the inhibition of NFκB and AP-1 and the activation of Nrf2 [42,43,44]. An in vitro study of microglial cells found that negative regulators of NFκB and AP-1, including Dusp1 and Trib3, were upregulated upon treatment with CBD [45]. However, CBD has also been shown to increase the expression of both TNFα and IκB in keratinocytes while simultaneously inhibiting the expression of pro-inflammatory NLRP3 and PGAM5, demonstrating that further study is needed to unravel the details of NFκB pathway modulation by CBD [42]. Multiple studies have demonstrated Nrf2 activation in CBD-treated keratinocytes and glioblastoma cells, including upregulation of the Nrf2 activators KAP1, p21, and p62 [42,45]. The effect of CBD on the production of reactive oxygen species (ROS) is also controversial. CBD has been reported to increase ROS in multiple models of cancer, including glioma, leukemia, breast cancer, and colorectal cancer, but CBD has also been reported to reduce oxidative stress in microglial cells exposed to LPS or keratinocytes exposed to UV radiation [42,46,47,48,49].

## 4. Pre-Clinical Studies: Role of CBD as a Component of Combination Therapies for Cancer

Several studies have shown that CBD can exert an anti-tumor effect in a broad range of preclinical cancer models by different mechanisms, including apoptosis, pyroptosis, the inhibition of angiogenesis, and inhibition of metastasis. The results of these studies are summarized in Table 1. CBD’s anti-cancer effects are mediated by multiple intracellular molecular changes, including the induction of gene/micro-RNA expression, protein phosphorylation, and ROS, as depicted in Figure 3. In human colorectal cancer cell lines (HCT-116, SW480, Caco-2), CBD reduced cell viability, with an IC50 value ranging from 4.7–20 µM [12]. Additionally, CBD caused cell cycle arrest by decreasing cyclin D3, CDK2, CDK4, and CDK6, with no impact on cell cycle inhibitor proteins p21 and p27. CBD induced colorectal cancer cell apoptosis by the activation of the ROS-mediated ER-stress pathway and its downstream regulators (p-eIF2 α, ATF3, ATF4, CHOP), which is reversible by adding the ROS scavenger, N-acetyl cysteine, or using CHOP small interfering RNA (siRNA) [12,49]. The observed cytotoxicity was mediated by the CB-2 receptor, as the pretreatment with CB-2 receptor inhibitor SR144528 reversed CBD-induced colorectal cancer cell apoptosis. Jeong et al. showed the involvement of ROS-mediated ER stress and NOXA in CBD-mediated cytotoxicity in HCT-116 and DLD-1 cells [49]. Furthermore, the anti-tumor activity of CBD was confirmed in an in vivo HCT-116 subcutaneous mouse model, and mediated by activating NOXA [49].

ROS also contributes to CBD-cytotoxicity in human leukemia cells by the upregulation of the NAD(P)H oxidases Nox4 and p22 (phox) and is reversed by ROS scavengers or NAD(P)H oxidase inhibitors [50]. Likewise, increased ROS was shown to mediate a therapeutic response in cell and mouse models of glioblastoma [46]. Microarray-based whole transcriptome gene expression analysis revealed that increased ROS led to a reduction in the self-renewal markers *Sox2*, *Id1*, and *STAT3* through activation of the p38 pathway partially mediated by NRF2. A similar RNA-seq transcriptomic analysis was performed on the neuroblastoma cell line SK-N-BE (2) following CBD treatment [51]. This analysis found that CBD promoted apoptosis through alterations to cholesterol synthesis, import, and trafficking. Cholesterol metabolism is known to contribute to the development of drug resistance in cancer; therefore, this finding suggests that the combination of CBD with other cancer therapies may help to prevent drug resistance [52]. Furthermore, a study of hepatocellular carcinoma cells treated with 40 µM CBD for 24 h found that CBD induced pyroptosis via the AKT pathway [53]. 

The access to raw transcriptomic data produced by these studies provided by the NCBI Gene Expression Omnibus (GEO) database enabled us to perform our own analysis aimed at the identification of genes and pathways commonly regulated by CBD in multiple types of cancer. Our analysis of the gene expression data provided by these three studies yielded 42 commonly differentially expressed genes (Appendix A). These genes include known mediators of CBD activity, *SQSTM1*, *GDF15*, and the metallothionein family [54,55,56] (Figure 4A). The analysis also revealed novel genes with no currently known association with CBD (PubMed search), including *ZFP36*, *PLK3*, and *TARS1*, which may be useful as novel targets for future studies. Major pathway hubs of CBD-altered genes were identified using Ingenuity Pathway Analysis (IPA, Qiagen), including the activation of p53, inhibition of AKT, the stress response transcription factors *ATF3/4*, and the DNA damage inducible transcript *DDIT3*. This finding is consistent with increased ROS, ER stress, and DNA damage. The gene set includes members of the p53, MAPK, PI3K, EGFR, oxidative stress, and apoptosis pathways (Figure 4B). Gene ontology analysis reveals an over-representation of genes regulating transcription/translation and coding for metabolite interconversion enzymes (Figure 4C). A broader analysis of prostate, breast, head and neck, and glioblastoma cancer cell lines also found that cholesterol synthesis, p53, cell cycle, and angiogenesis were among the most commonly targeted pathways by CBD [55]. This supports the hypothesis that CBD may have a pan-cancer activity.

Furthermore, in both ER-positive and ER-negative− breast cancer cell lines, CBD reduced cell viability in a dose-dependent manner, higher in CB-1- and CB-2-expressing T-47D cells [57]. Mechanistically, CBD inhibits EGF signaling in breast cancer cells and its downstream regulators, EGFR, AKT, NF-κB, MMP2, and MMP9 [58]. This anti-tumor mechanism was confirmed in 4T1.2 and MVT-1 mouse models, as CBD treated mice showed significantly lower tumor growth through the inhibition of ERK and AKT signaling. In a mouse model of colon cancer, tumors grown by injecting the CT26 cell line exhibited increased apoptosis and reduced expression of VEGF with CBD treatment [59]. In lung cancer cell lines A549 and H460, CBD reduced cell viability independently of CB-1, CB-2, and TRPV-1 receptors. However, the induction of apoptosis was dependent on increased COX-2, PPAR-γ, and prostaglandin levels after CBD treatment [41].

Indeed, the use of pharmacological inhibitors or siRNA to reduce PPAR-γ and COX-2 signaling reversed CBD-induced apoptosis in lung cancer cell lines and primary patient-derived lung tumor cells. Furthermore, CBD reduced the invasiveness of lung cancer cell lines, primary cells, and A549 xenograft mouse models by inducing ICAM-1 and TIMP-1, an effect dependent on CB-1, CB-2, TRPV-1 receptors, and p42/44 MAPK [60]. As a result, CBD treatment significantly reduced the number of metastatic lung nodules in a metastasis model. In gastric cancer cell lines, CBD caused G0-G1 cell cycle arrest and apoptosis through increased ROS production, cleaved caspases-3 and 9, decreased Bcl-2 levels, and mitochondrial dysfunction [13,61]. As a result of CBD-induced ER stress and mitochondrial dysfunction, Smac levels increased in the gastric cancer cells and in the MKN45 xenograft mouse model, causing the degradation of the anti-apoptotic protein XIAP. Similarly, CBD induced apoptosis in cervical and pancreatic cancer by increasing p53, caspase 3, and BAX, and inhibiting Kras-activated PAK-1 signaling [62,63].

Micro RNAs (miRNAs) also play a role in CBD-mediated cytotoxicity, as treated human neuroblastoma cells show reduced hsa-let-7a and increased has-mir-1972, which causes the increased expression of caspase-3 and decreased expression of BCL2L1 and SIRT2 genes, respectively [64]. RNA-seq analysis of CBD-treated head and neck squamous cell carcinoma HNSCC cells shows enrichment in processes involved in apoptosis, cell cycle arrest, and impaired DNA replication and repair [15]. Although CBD induces apoptosis in a broad range of cancers, it targets hepatocellular carcinoma (HCC) cells by inducing pyroptosis, as indicated by a balloon-like morphology; LDH release; and cleavage of caspase-3, PARP, and GSDME [53]. Additionally, CBD was able to reduce the glycolytic capacity of HCC cells by inducing IGFBP-1 and inhibiting the AKT/GSK3β. Marzeda et al. showed that CBD exerts a more potent cytotoxicity in melanoma cell lines (SK-MEL 28, A375, FM55P, and FM55M2) than in human keratinocytes [65].

Cancer treatment usually requires the use of combinatorial approaches due to the need for the targeting of specific oncogenic driver pathways, targeting cell survival mechanisms with chemotherapy, overcoming drug resistance mechanisms, and alleviating adverse effects. Several studies show that CBD, when used in combination with other anti-cancer agents, achieves synergy, or an additive effect, or antagonism. For example, with tamoxifen, CBD exerts an additive cytotoxicity in T-47D cells, whereas a triple combination of CBD, fulvestrant, and palbociclib was the most effective in reducing cell viability in both T-47D and MCF-7 cells [57]. In multiple myeloma cell lines, CBD was synergistic with bortezomib, inducing cell cycle arrest; ROS-dependent necrosis; and the inhibition of ERK, NF-kB, and AKT signaling [66]. In addition to its pro-apoptotic effects in human head and neck squamous cell carcinoma, CBD was synergistic with the chemotherapeutic agents: 5-FU, cisplatin, and Taxol [15]. In a mouse model of pancreatic ductal adenocarcinoma, the inhibition of GPR55 signaling by CBD was observed to increase the effectiveness of gemcitabine and prolong survival [67]. Although CBD showed an additive cytotoxicity when added to mitoxantrone in melanoma cell lines, it was antagonistic to cisplatin treatment in SK-MEL 28, A375, and FM55P cell lines. The cell-line-dependent effect of the addition of CBD to cisplatin treatment (synergism vs. antagonism) will be critical to understanding CBD’s translation to clinical use. Cisplatin is commonly used to treat multiple types of cancer, and, therefore, molecular or pathological markers will need to be discovered to indicate when CBD should be added to cisplatin and when it should be avoided.

**Table 1 ijms-23-12956-t001:** Preclinical studies of CBD in cancer.

Summary of Preclinical Studies Involving CBD Treatment of Cancer Cells and Mouse Tumors			
Preclinical Systems	Pathways	Molecules and Mechanisms Involved	Ref.
Cell Culture Models			
Colorectal cancer cells: HCT-116, SW480/620, Caco-2	Cell cycle arrest	cyclin D3, CDK2, CDK4, and CDK6	[12]
	Apoptosis-ROS-ER stress pathway	p-eIF2 α, ATF3, ATF4, CHOP	[49]
	EMT	WNT/β-catenin	[68]
Leukemia cells	ROS	NADPH oxidases Nox4, p22-phox	[50]
Glioblastoma cells	ROS	SoX2, ID1, STAT3	[46]
	ROS/apoptosis	Cholesterol	[55]
Hepatocellular carcinoma cells: HCC	Pyroptosis	AKT pathway: inducing IGFBP-1 and inhibiting the AKT/GSK3β LDH release and cleavage of caspase-3, PARP, and GSDME	[65]
Lung cells: A549, H460, and primary lung tumor cells	Apoptosis	COX-2, PPAR-γ, and prostaglandin	[41]
Breast (ER+ and ER−): T-47D	EGF signaling	EGFR, AKT, NF-κB, MMP2, and MMP9	[58]
Gastric cells	Increased ROS cell cycle arrest, apoptosis	Cleaved caspases-3 and 9, decreased Bcl-2 levels and mitochondrial dysfunction	[13,61]
Cervical/Pancreatic cells	Apoptosis	p53, caspase 3, BAX, and inhibiting KRAS-activated PAK-1 signaling	[62,63]
Neuroblastoma cells	Apoptosis	hsa-let-7a, has-mir-1972: caspase-3(UR) and BCL2L1 and SIRT2 (DR)	[51,64]
Animal Models			
Colorectal cancer HCT-116 subcutaneous model	Anti-tumor	ROS, NOXA activation	[49]
Mouse model of glioblastoma	Anti-tumor	ROS	[46]
Colon: CT26	Apoptosis	VEGF	[59]
Lung: A549 xenograft mouse model	Anti-tumor	ICAM-1 and TIMP-1, p42/44 MAPK	[60]

In U87MG human glioblastoma cells, CBD increased the cytotoxic effect of γ-irradiation; however, it also increased the activation of ATM kinase, an important player in DNA repair [69]. This increase could be overcome by the addition of an ATM kinase inhibitor. Furthermore, CBD was recently shown to inhibit SARS-CoV-2 replication through the activation of ER stress and innate immune responses [70]. Although the effects of CBD on the tumor-immune microenvironment in vivo require further study, CBD’s modulation of the immune response may also be advantageous in cancer therapy, as lymphoid tissues are known to express high levels of CBD receptors (Figure 1).

CBD and THC are present in the cannabis plant, and they are frequently used together for the treatment of human diseases. Indeed, the majority of clinical studies of CBD in cancer patients have included THC (Table 2). Therefore, it is vital to understand the similar and disparate effects of these compounds on cancer cells. THC has been reported to have anti-cancer properties in cell culture models of lung, urothelial, and breast cancer [71,72,73]. In lung cancer, THC inhibited proliferation by reducing the activity of the EGFR pathway, including ERK, JNK, and AKT [71]. THC also inhibited the ability of cancer-associated fibroblasts to support lung cancer cell growth and epithelial-to-mesenchymal transition [74]. THC and CBD were shown to act synergistically, reducing cell migration and increasing apoptosis through the CB-1/2 receptors in urothelial carcinoma [72]. In breast cancer, THC reduced cell proliferation by activating JunD and the downstream CDK inhibitor, p27 [73]. This study performed gene expression analysis of THC-treated EVSA-T cells by microarray (Figure 5A, Appendix A), enabling the comparison of the gene expression profile elicited by THC to that elicited by CBD (shown in Figure 4). Only three genes were found to be commonly differentially expressed in both CBD- and THC-treated cells: asparagine synthetase (*ASNS*), homocysteine inducible ER protein with ubiquitin-like domain 1 (*HERPUD1*), and solute carrier family 3 member 2 (*SLC3A2*). These were all upregulated in both treatments. The THC gene set includes members of the p53, Wnt, TGF-β, interleukin signaling, and apoptosis pathways (Figure 5B). Gene ontology analysis reveals an over-representation of genes coding for both metabolic- and protein-modifying enzymes along with protein activity regulators and RNA metabolism proteins (Figure 5C). Despite common reductions in cell proliferation seen with both compounds, the gene expression patterns shown in Figure 4 and Figure 5 are quite different. Where CBD was predicted to activate p53 and inhibit AKT, THC was predicted to inhibit p53, and resulted in the upregulation of Myc. This analysis suggests that the combination of THC and CBD may not always be advantageous if the two compounds initiate opposing downstream signaling effects. Indeed, comparing the two data sets at the pathway level using IPA revealed that CBD is predicted to activate apoptosis and inhibit proliferation, whereas THC is not (Figure 6). However, it must be considered that the THC expression profile analyzed herein is only based on a single dataset utilizing only the EVSA-T cell line. Therefore, more work is needed to rigorously characterize the downstream effects of THC in various cancer types containing differing driver mutations. Furthermore, THC was demonstrated to enhance breast cancer growth in the 4T1 mouse model by inhibiting the anti-tumor Th1 immune response through increased levels of IL-4 and IL-10, and to increase the growth of colon cancer by promoting angiogenesis in the HCT-116 xenograft model [75,76]. Though the molecular effects of cannabinoids on cancer cells are still not fully understood, THC is known to influence inflammation, cytokine production, and reduce TNFα [77]. Taken together, these results point to the possibility of greater benefits with CBD in the absence of THC; however, the translational significance of these pre-clinical studies remains ambiguous.

**Table 2 ijms-23-12956-t002:** Clinical studies of CBD in cancer patients.

Trial (N)-Design	Intervention	Results	Ref.
NCT00530764 (N = 360) A Study of Sativex^®^ for Pain Relief in Patients with Advanced Malignancy (SPRAY). Duration: 9 weeks Design: Double blind, randomized, placebo-controlled, parallel group, dose-range exploration study. Results Posted: 17 June 2011	Experimental: Sativex Each 100 μL oromucosal spray actuation delivered: (2.7 mg THC + 2.5 mg CBD) Low Dose: Range of 1 to 4 sprays per day. Maximum daily dose: 10.8 mg THC and 10 mg CBD. Medium Dose: Range of 6 to 10 sprays per day. Maximum daily dose: 27 mg THC and 25 mg CBD. High Dose: Range of 11 to 16 sprays per day. Maximum daily dose: 43.2 mg THC and 40 mg CBD. Placebo: 1–16 sprays per day	-Number of patients with at least 30% improvement in Numerical Rating Scale (NRS) average pain score from baseline was not significant with any dose. -Change in cumulative average pain response was significant in low and med doses. -Change in sleep disruption NRS was significant in low dose. -No change in Montgomery Asberg Depression Rating Scale (MADRS).	[78]
NCT00675948 (N = 43) Study to Compare the Safety and Tolerability of Sativex^®^ in Patients with Cancer-Related Pain. Duration: 2 weeks Design: multicenter, open-label, self-titrated dose Results Posted: 13 September 2012	Sativex^®^ or GW-2000-02 oromucosal spray (self-administered, self-titrated). Sativex: THC 27 mg/mL + CBD 25 mg/mL Maximum tolerated/allowable dose was 130 mg THC and 120 mg CBD in each 24 h period. GW-2000-02: THC 27 mg/mL. Maximum tolerated/allowable dose was 130 mg THC in each 24 h period.	-Change from baseline in mean Brief Pain Inventory-Short Form scores for “pain severity” and “worst pain” decreased (improvement) at each visit in the THC/CBD spray group. -The EORTC Quality of Life Questionnaire (C30 scores) decreased (improvement) from baseline for insomnia, pain, and fatigue. -No new safety concerns associated with the extended use of THC/CBD spray.	[79]
NCT01337089 (N = 660) Long-Term Safety of Sativex as Adjunctive Therapy in Patients with Uncontrolled Persistent Chronic Cancer-Related Pain. Duration: 6 months Design: multicenter, non-comparative, open label extension Results Posted: 23 April 2018	Nabiximols (self-administered, self-titrated) oromucosal spray. Maximum of 10 sprays per day for 6 months. Nabiximols contained: THC 27 mg/mL CBD 25 mg/mL Each 100 μL actuation delivered: 2.7 mg THC 2.5 mg CBD	-No change from baseline in mean NRS average pain score. -No change from baseline in mean sleep disruption NRS. -Overall patient satisfaction was high. # of patients extremely satisfied: 56 (9.1%); very satisfied: 230 (37.2%); slightly satisfied: 185 (29.9%); neutral: 82 (13.3%).	[80]
NCT01361607 (N = 399) Sativex^®^ for Relieving Persistent Pain in Patients with Advanced Cancer (SPRAY III). Duration: 9 weeks Design: multi-center, double-blind, randomized, placebo-controlled. Results Posted: 23 April 2018	Experimental: Nabiximols (self-administered, self-titrated) oromucosal spray. Minimum 3 and maximum of 10 sprays per day for 5 weeks. Nabiximols contained: THC 27 mg/mL CBD 25 mg/mL Each 100 μL actuation delivered: 2.7 mg THC 2.5 mg CBD Placebo (GA-0034): Placebo (self-administered) 100 μL oromucosal spray. Maximum of 10 sprays per day for 5 weeks. Placebo contained ethanol: propylene glycol (50:50)	-Percent improvement from baseline in mean NRS average pain score: Ex. 7.2% (0.0 to 30.0) Placebo 9.5% (−2.9 to 25.7) -Change from baseline in mean sleep disruption NRS: Ex. −0.9 Placebo −1.1 -Change from baseline in daily total opioid use (morphine equivalent): Ex. −6.5 Placebo 2.3 -Change from baseline in daily breakthrough opioid dose (morphine equivalent): Ex. −4.4 Placebo 0.5	[81]
NCT01262651 (N = 397) Sativex^®^ for Relieving Persistent Pain in Participants with Advanced Cancer Duration: 9 weeks Design: multi-center, double-blind, randomized, placebo-controlled Results Posted: 23 April 2018	Nabiximols (self-administered, self-titrated) oromucosal spray. Minimum 3 and maximum of 10 sprays per day for 5 weeks. Nabiximols contained: THC 27 mg/mL + CBD 25 mg/mL Each 100 μL actuation delivered: 2.7 mg THC + 2.5 mg CBD Placebo (GA-0034): Placebo (self-administered) 100 μL oromucosal spray. Maximum of 10 sprays per day for 5 weeks. Placebo contained ethanol: propylene glycol (50:50).	-Median percent improvements from baseline in average pain NRS score: Intention-to-treat population: Nabiximols: 10.7% (*p* = 0.0854), Placebo: 4.5% -Per-protocol population: Nabiximols: 15.5% (*p* = 0.0378), Placebo: 6.3% “Nabiximols was statistically superior to placebo on two of three quality-of-life instruments at Week 3 and on all three at Week 5. The safety profile of Nabiximols was consistent with earlier studies.”	[78,82]
NCT00674609 (N = 177) A Study of Sativex^®^ for Pain Relief in Patients with Advanced Malignancy (SPRAY) Duration: 2 weeks Design: multicenter, double blind, randomized, placebo-controlled, parallel group study to evaluate the efficacy of Sativex^®^ and GW-2000-02 Results Posted: 13 August 2012	Sativex^®^ (self-administered, self-titrated) oromucosal spray (100 µL/actuation). THC 27 mg/mL CBD 25 mg/mL Maximum tolerated/allowable dose was 130 mg THC and 120 mg CBD in each 24 h period. GW-1000-02: (THC Alone) (self-administered, self-titrated) THC 27 mg/mL. Maximum tolerated/allowable dose was 130 mg THC in each 24 h period. Placebo contained colorants and excipients.	-Change from baseline in mean pain NRS score was statistically significant for Sativex, but not GW-1000-02. -Patients with a reduction of more than 30% from baseline pain NRS score. -No change from baseline in median dose of opioid background medication or mean number of doses of breakthrough medication across treatment groups. -No significant group differences were found in the NRS sleep quality or nausea scores or the pain control assessment.	[83]
NCT01424566 (N = 406) A Two-Part Study of Sativex^®^ Oromucosal Spray for Relieving Uncontrolled Persistent Pain in Patients with Advanced Cancer Duration: 11 weeks Design: multi-center; placebo-controlled; aimed to determine the efficacy, safety, and tolerability of Nabiximols. Results Posted: 23 April 2018	Nabiximols (self-administered, self-titrated) oromucosal spray. Minimum 3 and maximum of 10 sprays per day for 2 or 7 weeks. Nabiximols contained: THC 27 mg/mL CBD 25 mg/mL With excipients Each 100 μL actuation delivered: 2.7 mg THC + 2.5 mg CBD Placebo was self-administered by participants as a 100 μL oromucosal spray up to a maximum of 10 sprays per day for 5 weeks. Placebo oromucosal spray contained ethanol:propylene glycol (50:50) with excipients.	-Overall percent improvement or mean change in pain NRS score was not statistically significant. -Statistically significant treatment effect for Sativex was identified in US patients <65 years (*p* = 0.040). “Treatment effect in favor of Sativex was observed on quality-of-life questionnaires, despite the fact that similar effects were not observed on NRS score. The safety profile of Sativex was consistent with earlier studies, and no evidence of abuse or misuse was identified.”	[81]
EudraCT trial no. 2014-005553-39 (N = 23): Effects of Sativex on Blood Leukocytes in Patients with Lymphoma/Chronic Lymphocytic Leukemia Duration: Single dose Design: Therapeutic exploratory (Phase II), open label, not placebo-controlled Results Posted: 17 January 2022	Asymptomatic patients with CLL or marginal zone lymphomas received a single dose of THC/CBD, starting from 2.7 mg THC and 2.5 mg CBD (one actuation of Sativex) to 18.9 mg THC and 17.5 mg CBD.	-A significant reduction in leukemic B cells (median, 11%) occurred in the blood within two hours (*p* = 0.014), and remained for 6 h without induction of apoptosis or proliferation. All effects were gone by 24 h. Normal non-leukemic B cells and T cells were also reduced.	[84]

Abbreviations: Ex, Experimental; EORTC, European Organization for Research and Treatment; MADRS, Montgomery-Asberg Depression Rating Scale; NRS, Numerical Rating Scale.

## 5. Clinical Studies of CBD in Cancer

The completed clinical studies of CBD in cancer either for symptom palliation or cancer treatment are outlined in Table 2. A total of seven studies of CBD in cancer patients have posted results to the NIH’s clinicaltrials.gov database at the time of this writing. The majority of studies to date have focused on pain and opioid use reduction as a primary endpoint, with secondary endpoints including anxiety, sleep, and various quality-of-life measurements. The majority of these studies have used combinations of THC and CBD; therefore, further work is needed to characterize the potential of CBD treatment either alone or in combination with chemo- or targeted therapies without THC. Pharmacokinetic studies of CBD in humans have demonstrated its half-life to be between 1.4 and 10.9 h when delivered by oromucosal spray or 1–2 h when delivered orally [85,86]. The plasma clearance of CBD is affected by the fed vs. fasted state, with the plasma clearance rate increasing 5–10-fold when fasted. The C_max_ and AUC are dose-dependent, with a 10 mg oral dose resulting in a C_max_ of 2.5 ng/mL, whereas an 800 mg oral dose resulted in a C_max_ of 221 ng/mL at T = 3 h. C_max_ and AUC have also been reported to increase when fed. Higher C_max_ can be achieved by IV delivery, where an IV dose of 20 mg resulted in a C_max_ of 686 ng/mL at T = 3 min; however, CBD delivered by IV is cleared more quickly, resulting in a plasma concentration of just 48 ng/mL at T = 1 h [86].

Initial pilot studies such as the open-label trial conducted by Good et al. showed CBD to be well tolerated (at 300 mg/day), and revealed a trend of decreasing total symptom distress score after completing the 14-day regimen (compared to baseline) [9]. A subsequent dose range exploration study of a combination CBD/THC oromucosal spray in cancer patients tested doses ranging from 2.5 + 2.7–40 + 43.2 mg, and also found that these doses were well tolerated, with the exception of the high dose group, in which only 66% of patients were able to complete the study at this dose [78]. The average pain score was significantly reduced in the low and medium doses compared to the placebo, whereas changes in sleep disruption were significantly improved in the low dose and trended towards significance in the medium dose. Following the determination of an appropriate dose range, a study was performed to assess the potential use of Sativex (CBD+THC) for pain relief in patients with advanced malignancy [83]. This study was randomized, double-blind, and placebo-controlled, although the method of randomization was not specified. It enrolled 177 participants and lasted for 2 weeks. Patients either used Sativex at a maximum dose of 120 mg CBD+ 130 mg THC per day, THC alone at a maximum dose of 130 mg per day, or a placebo. The treatments were delivered as an oromucosal spray. The study found a significant reduction from baseline pain numerical rating scale (NRS) score in the patients taking CBD+THC, but not THC alone or placebo. Of the patients taking CBD+THC, 43% of reported a greater than 30% reduction in pain compared to 23% of the patients taking THC or 21% taking a placebo. This study found no change in median opioid dose, sleep quality, or nausea scores between the groups. A follow-up study allowed patients to continue taking the study medication for an additional 2 weeks in order to better assess the safety and tolerability of CBD+THC and determine its effectiveness to alleviate cancer-related pain [79]. This study was open-label with a self-titrated dose and 43 participants. Patients taking CBD+THC experienced improvements in Brief Pain Inventory-Short Form scores for “pain severity” and “worst pain” and in the European Organization for Research and Treatment of Cancer Quality of Life Questionnaire (C30 scores). Evaluation scores in the THC-alone group were not significantly changed, and no new safety concerns of either treatment were noted. These results indicate that the CBD component of THC/CBD treatments is critical for pain reduction, and future studies with the addition of a CBD-only group are needed to fully understand the relationship between THC and CBD in cancer pain reduction.

The results of four additional trials extending the scope of these earlier studies of CBD+THC in cancer-related pain were published in 2018. The first extended the treatment duration to 6 months at the same dose used previously and enrolled 660 patients with an open-label design [80]. This study was non-comparative and not placebo-controlled, as the only study medication was the CBD+THC combination. No change was observed in pain or sleep disruption NRS score from baseline to study completion; however, the majority of patients were either slightly, very, or extremely satisfied with the treatment. The second trial enrolled 399 patients for a 9-week double-blind, randomized, placebo-controlled study of the effectiveness of CBD+THC at a maximum dose of 25 mg CBD + 27 mg THC per day [81]. Patients taking CBD+THC experienced a 7.2% improvement in pain NRS score over the course of the study, whereas patients taking a placebo experienced a 9.5% improvement. However, patients taking CBD+THC reduced their daily total opioid use on average over the course of the study, whereas patients taking a placebo increased their opioid use. Though the reduction of opioid use may be an important measure for determining the success of CBD in pain treatment, opioid use should be tested in a standalone group, since allowing different levels of opioid use in the CBD+THC vs. control groups will inevitably confound measurements of pain. The third trial focused selectively on patients with advanced cancer experiencing persistent pain [82]. It enrolled 397 patients for a 9-week double-blind, randomized, placebo-controlled trial. CBD+THC was delivered as an oromucosal spray with the same maximum dose as the previous study. Patients taking CBD+THC experienced a 10.7% improvement in pain NRS score over the course of the study, whereas patients taking a placebo experienced a 4.5% improvement. Patients taking CBD+THC also reported significant improvements in quality-of-life questionnaires. Finally, the fourth study enrolled patients with uncontrolled persistent pain due to advanced cancer [81]. The placebo-controlled study had 406 participants and lasted for 11 weeks. Again, the same maximum dose of 25 mg CBD + 27 mg THC was used. An overall improvement in the pain NRS score was not found to be significant when all patients (U.S. and European) were included in the analysis. However, a significant NRS score reduction was identified in U.S. patients younger than 65 years of age. A positive effect was also observed in quality-of-life questionnaires. This result suggests that age may be a factor in CBD’s ability to reduce pain; however, the effect of age is yet to be corroborated by additional trials. The different results observed in U.S. vs. European patients highlight how variables which may be different between the two study groups (additional pain medications, type of cancer treatment, diet, exercise, co-morbidities) could influence the outcome of CBD treatment. Future studies are, therefore, necessary to understand the influence of these variables on CBD effectiveness.

In addition to clinical studies of pain, one single-dose, open-label, exploratory study was performed to evaluate the effects of CBD+THC (2.5 + 2.7 mg–17.5 + 18.9 mg) on the blood leukocytes in patients with lymphoma or chronic lymphocytic leukemia [84]. The study found a significant reduction in B cell counts 2–6 h post-treatment; however, no change in apoptosis or proliferation was observed, and the effect did not last 24 h. The effect was not cancer-specific, as normal B cell and T cell counts were also reduced. Overall, significant benefits, including patient satisfaction, quality of life, reduced sleep disruption, reduced pain, and reduced opioid use, have all been demonstrated in clinical trials of cancer patients through the use of CBD or CBD+THC. The trials completed so far enrolled patients regardless of cancer type with the exception of the leukemia study. The most common cancer types enrolled were breast, lung, colon, and prostate [78,81]. Future studies will be needed to elucidate the potential benefits of CBD treatment within specific cancer types, since cancer-type-specific post-hoc analysis has not yet been performed. Furthermore, the reduction in pain was not observed consistently in all studies. The dose range exploration study found the best improvement in pain at low and medium doses, with no improvement at the high dose [78]. Though the cause of this lack of positively correlated dose response is not clear, it may be associated with the increased rates of nausea, dizziness, and vomiting observed in the high-dose group. It remains to be seen if CBD alone elicits the same effect on pain and the same adverse events at this dose, since this study only tested CBD+THC. Furthermore, several of the studies outlined in Table 2 allowed patients to alter their own dose, which makes results more difficult to interpret, since the doses were not truly randomized by this design. These inconsistencies highlight the need for additional well-controlled and randomized trials, more investigation into the mechanisms by which CBD can treat pain, and the identification of clinical biomarkers or criteria to indicate which specific patients may benefit most from CBD.

## 6. CBD Use in Rehabilitation of Cancer Patients

Cancer patients commonly experience both physical and psychological symptoms, including pain, nausea, sleep disturbance, fatigue, anxiety, and depression [87,88]. These symptoms are most commonly managed using medications (NSAIDs, opioids, anti-emetics, stimulants) and/or cognitive behavioral therapy [88]. Despite receiving these treatments, more than half of cancer patients report challenges with activities required for daily living, and 20–25% of patients receiving opioids for chronic pain do not achieve a greater than 30% reduction in pain [88,89]. Furthermore, opioids have their own detrimental side effects, including constipation, sedation, loss of concentration, depression, and dependency [87,90]. Therefore, novel therapies are desperately needed to alleviate cancer symptoms and to reduce the reliance on opioids for analgesia in cancer patients.

CBD is commonly used as an over-the-counter supplement for multiple conditions, including sleep disorders, anxiety, and pain [91,92]. As of 10/1/2022 there are a total of 411 clinical studies of CBD either completed or ongoing listed on the NIH’s clinicaltrilas.gov database. Sixty of these studies have posted results thus far [93]. There are 96 studies to treat pain, 16 involving sleep disorders, 4 for appetite, 31 for anxiety/depression, 5 for nausea, and 5 which include a measure for quality of life. Novel treatments for these conditions are of great clinical interest to cancer patients, as they are common symptoms of cancer or side effects of cancer treatments. Therefore, the evidence of CBD’s effectiveness to treat these symptoms caused by other conditions also supports its potential for use in cancer and the need for future studies to standardize the indications for its use, including dose, route, and combinations with cancer drugs.

## 7. Pain

The most well-studied indication for CBD in cancer is pain reduction or improved pain management (reduced opioid use). In patients with pain due to spinal cord injury, oral THC/CBD self-administered up to 130 mg/120 mg significantly increased the reduction from baseline in the mean brief pain inventory score at the end of treatment [94]. Patients were also significantly more likely to rate their condition as “very much improved, much improved, or minimally improved” compared to placebo. Furthermore, a study of 94 chronic pain patients revealed that adding 15–60 mg of CBD-rich hemp extract capsules per day to their current treatments enabled 53% of patients to reduce their opioid medications by the completion of the 8-week study [95]. These patients also exhibited a significant reduction in the assessment of pain intensity and interference at the conclusion of the study compared to the baseline. On the other hand, a study using the cold pressor test found no decrease in pain tolerance or threshold in healthy volunteers using a single dose of 200–800 mg CBD compared to a placebo [96]. Furthermore, a placebo-controlled trial of 20–30 mg daily CBD in hand osteoarthritis patients found no significant change in pain intensity [97]. These findings illustrate that the analgesic effects of CBD may be limited to certain types of pain at certain doses, and that CBD’s beneficial effects on mood and anxiety may also influence some measures of pain perception.

## 8. Sleep

In a study of Parkinson’s disease patients (N = 10), CBD started at 5 mg/kg and increased to 20 mg/kg over 5 weeks was observed to lower the Scales for Outcomes in Parkinson’s Disease (SCOPA) Sleep-night Time Sleep score (0–18) by an average of 2.8 points [98] (a lower score is better). In a study of Veterans (N = 80) with post-traumatic stress disorder (PTSD), smoking cannabis containing either high THC/low CBD or high CBD/low THC increased “sleep efficiency” (proportion of sleep period actually spent asleep) compared to placebo cannabis [99]. THC/CBD taken as an oromucosal spray also significantly improved sleep quality compared to baseline in multiple sclerosis patients [100]. Chronic pain patients taking CBD-rich hemp extract capsules not only benefited by reducing opioid usage, but also showed a significant improvement in their Pittsburg Sleep Quality Index (PSQI) [95].

## 9. Anxiety/Depression

Psychological symptoms including anxiety, depression, stress, fatigue, and mood changes are well known to negatively affect the quality of life in cancer patients [101,102]. As noted above, cancer patients participating in clinical trials of CBD or CBD/THC have reported improvements in assessments of these symptoms. Furthermore, trials of CBD in diseases other than cancer have also provided evidence of its potential to treat these psychological symptoms. Acute CBD administration at a dose of 300 mg decreased anxiety (heart rate, blood pressure, and Visual Analog Mood Scales) in patients with Parkinson’s disease during a simulated public speaking test [103]. A trial of 31 people with treatment-resistant anxiety disorders found that Overall Anxiety Severity and Impairment Scale scores improved significantly with 12-week CBD treatment on a flexible schedule, up to 800 mg per day [104]. Symptoms of depression were also reduced significantly.

## 10. Concluding Remarks

CBD has great potential to improve the lives of cancer patients both by alleviating the symptoms of pain, sleep disturbance, and anxiety, but also by synergistic activity with anti-cancer treatments to reverse or eliminate the growth of tumors causing these symptoms. Pre-clinical evidence in cell and mouse models supports the use of CBD as an anti-cancer therapy; however, clinical knowledge is currently lacking in this area. The effectiveness of CBD has been demonstrated in models of lung, breast, and colon cancer, as well as leukemia and glioblastoma. CBD has been shown to be toxic to cancer cells in vitro, and it is also generally well tolerated in the clinic. Furthermore, synergistic activity has been reported between CBD and several cancer drugs in vitro, including the DNA replication inhibitor cisplatin, the proteasome inhibitor bortezomib, and the microtubule stabilizer paclitaxel. Clinical trials so far have reported significant reductions in both pain and opioid use in cancer patients taking CBD/THC, which could not be matched by THC alone. Though this result is encouraging, these trials lack a CBD intervention without THC, and also lack true randomization (due to self-titration of dose). Patients with different types of cancer are reasonably well distributed between CBD/THC vs. placebo groups, but the effect of cancer type on the outcome of CBD/THC treatment is not adequately studied. Because of these limitations in the current data regarding CBD’s effectiveness to treat pain, future studies are needed to rigorously define variables influencing its analgesic effect, including age, cancer type, additional pain medications prescribed, type of pain experienced by the patient, co-morbidities, etc. More focused studies of individual cancer types could compare the effectiveness of CBD in patients who receive different treatments (surgery; radiation; chemo-, targeted-, and immuno-therapies).

Interest in the use of CBD to treat cancer is currently high among researchers and clinicians, as indicated by the large number of recently published articles. Several recent reviews of the topic have examined relevant aspects, including the biochemical, pharmacological, and molecular mechanisms of CBD activity in cells [4,23]; CBD’s mechanism-of-action in cell and mouse models of cancer treatment [105,106]; and its clinical use to manage cancer symptoms [107,108]. We have aimed to synthesize this information and identify unmet needs in the field (standard dosing and true randomization, comparison between pure CBD and CBD+THC formulations, and impact of drug combinations on CBD efficacy), which could serve as the focus for future research and, thus, hasten the clinical translation of CBD. Towards this goal, our own meta-analysis of in vitro differential gene expression studies presented herein suggests that although CBD and THC share an affinity for the CB receptors, their effects on key cellular processes in cancer, including apoptosis, proliferation, and the metabolism of ROS, can be quite different. Based on this analysis, CBD appears to have greater potential to induce apoptosis and inhibit the proliferation of cancer cells than THC. However, more rigorous in vitro studies are needed to better define these differences between CBD and THC treatment at the molecular level. Nevertheless, this finding highlights the need for clinical studies of the anti-cancer properties of CBD or CBD combination therapies using formulations that do not include THC. Future studies of CBD formulations without THC are also needed to better understand the role of CBD alone in alleviating cancer symptoms.

## Figures and Tables

**Figure 1 ijms-23-12956-f001:**
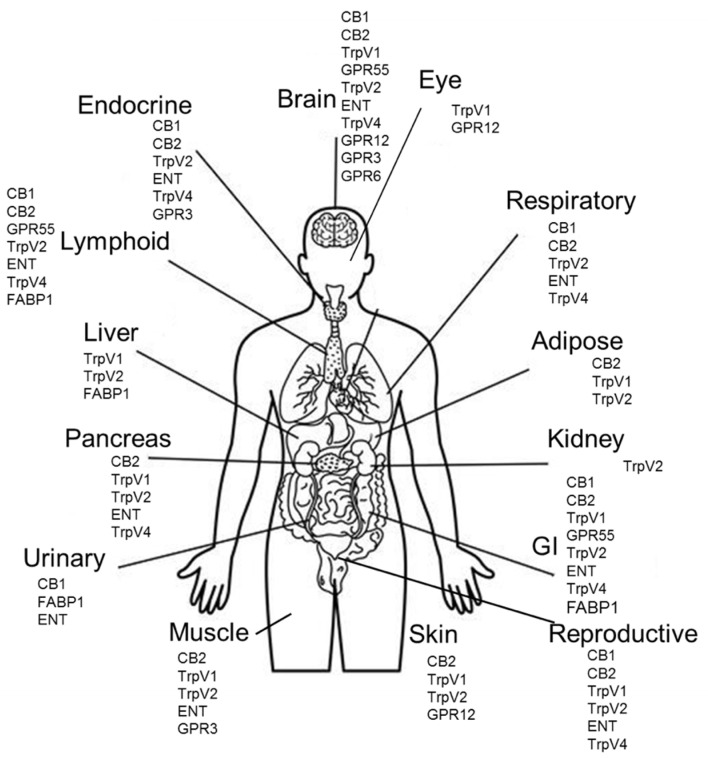
Physiological locations of CBD receptors. CBD receptor protein expression by organ. Proteins with greater than 10 normalized transcripts per million (nTPM) RNA expression or medium/high protein expression score according to the Human Protein Atlas (proteinatlas.org) are listed for each organ.

**Figure 2 ijms-23-12956-f002:**
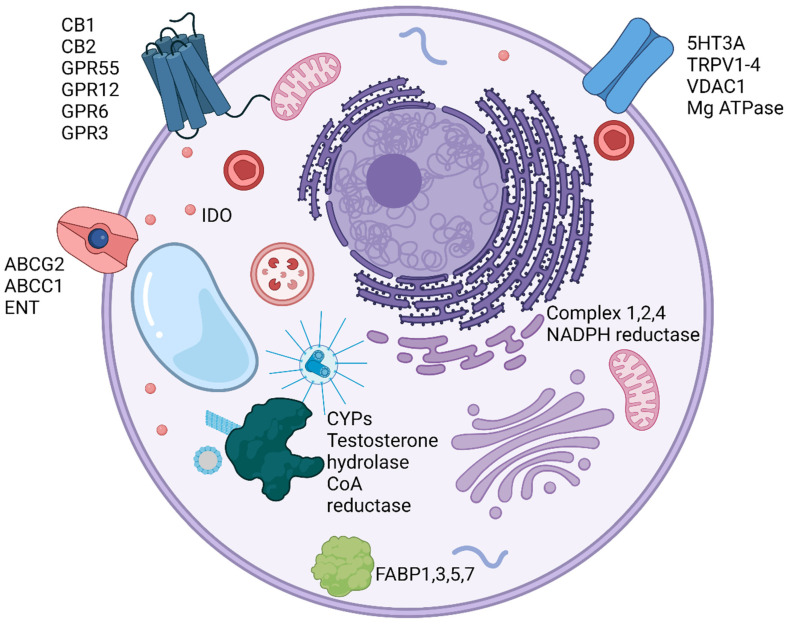
Molecular targets of CBD. Cellular proteins known to directly interact with CBD are shown. (Created with BioRender.com).

**Figure 3 ijms-23-12956-f003:**
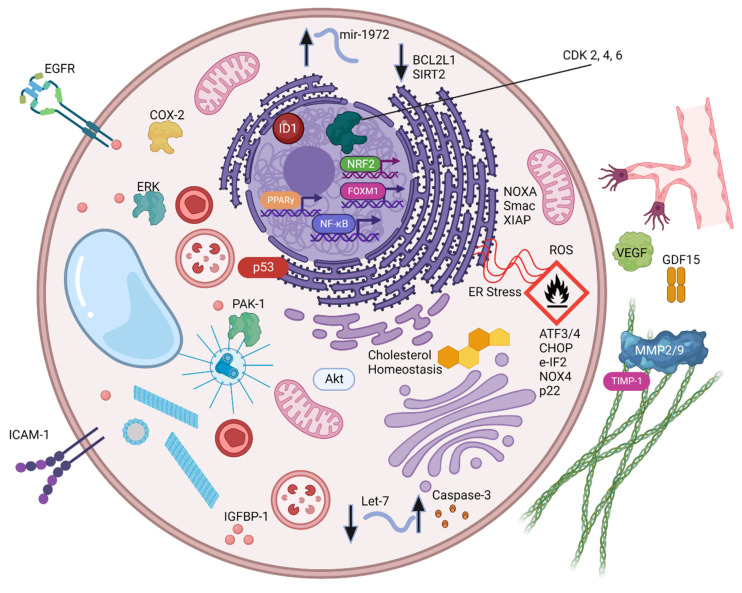
Evidence for CBD combination therapies for cancer. Molecular targets reported to be responsible for the anti-cancer effects of CBD either when used alone or in combination with known cancer therapies are shown. (Created with BioRender.com).

**Figure 4 ijms-23-12956-f004:**
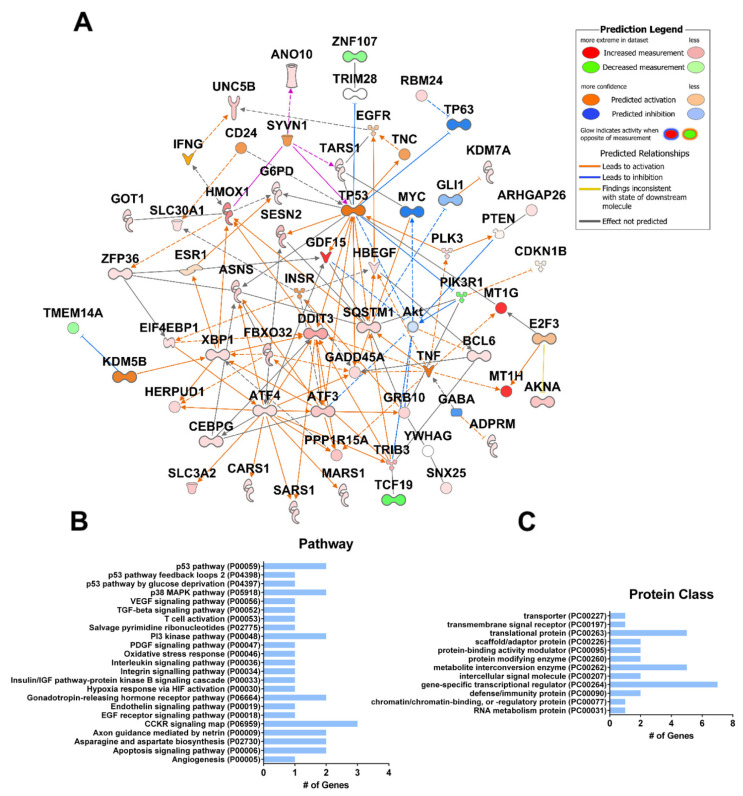
Common gene expression changes in CBD-treated SK-N-BE (2) neuroblastoma cells, hepatocellular carcinoma cells, and primary glioma stem cells. (**A**) Gene expression data were obtained from the NCBI GEO Database Series GSE151512 for SK-N-BE(2) cells, Series GSE179661 for hepatocellular carcinoma (HepG2 and MHCC97H), and Series GSE57978 for glioma stem cells. SK-N-BE (2) cells were treated with 20 µM CBD or vehicle (ethanol) for 24 h (n = 4). Three primary glioma stem cell lines were cultured as neurospheres and treated with 2 µM CBD or vehicle (ethanol) for 48 h (n = 3). Hepatocellular carcinoma cells were treated with 40 µM CBD for 24 h (n = 2). Significantly differentially expressed genes in each dataset were compared (*p* ≤ 0.05). Ingenuity Pathway Analysis (IPA Qiagen) was used to create a network of known interactions between the differentially expressed genes common to both studies. The average fold change of common differentially expressed genes is shown. (**B**) Functional gene ontology classification of genes shown in (**A**) according to their gene ontology “Pathway” or (**C**) “Protein class” terms (pantherdb.org). # = number.

**Figure 5 ijms-23-12956-f005:**
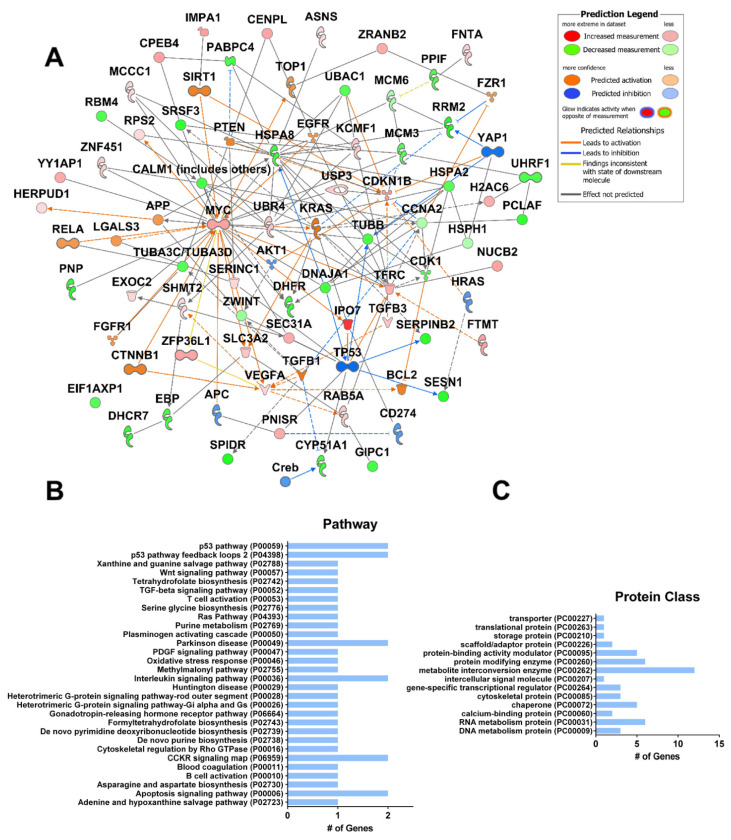
Gene expression changes in breast cancer cells (EVSA-T) treated with THC. (**A**). Cells were treated with 3 µM or 5 µM THC for 8 h or 24 h. Genes which were modified irrespective of time or concentration are shown (Series GSE8502 n = 4). Gene expression was assayed using Spanish National Cancer Research Center (CNIO) Oncochip containing 6386 genes. IPA (Qiagen) was used to create a network of known interactions between the differentially expressed genes. (**B**). Functional gene ontology classification of genes shown in (**A**) according to their gene ontology “Pathway” or (**C**) “Protein class” terms (pantherdb.org). # = number.

**Figure 6 ijms-23-12956-f006:**
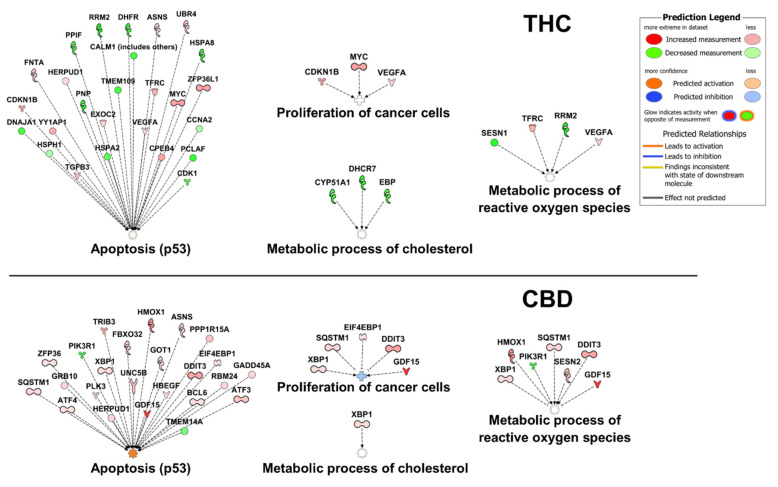
Pathway and function analysis of THC vs. CBD treatment induced differentially expressed genes in cancer. Differentially expressed genes in cancer cells treated with THC (Appendix A) or CBD (Appendix A) were analyzed for their contribution to common cellular functions in cancer using IPA (Qiagen). Genes identified by the IPA database as members of the “apoptosis”, “proliferation of cancer cells”, “metabolic process of cholesterol”, or “metabolic process of reactive oxygen species” pathways are shown for THC (**top**) and CBD (**bottom**).

## Data Availability

Datasets analyzed in this article can be found at: NCBI Gene Expression Omnibus (GEO) Database, Series GSE151512, GSE179661, GSE57978, and GSE8502.

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
