# Peer review of "Role of Cannabidiol for Improvement of the Quality of Life in Cancer Patients: Potential and Challenges"

_ijms, 2022, doi:10.3390/ijms232112956_

Round 1
Reviewer 1 Report
Dear authors,
The topic of this article is interesting. The information about "Biology and Molecular Targets of CBD" and "Role of CBD in Cancer treatment, pre-Clinical studies: CBD as a Component of Com-112 bination Therapies for Cancer" were presented followed by "Clinical Studies of CBD in Cancer" like the literature report. There was no critical evaluation of that information.
Therefore, the critical evaluation should be added to demonstrate the knowledge and understanding of the authors. The reader will get the advantage from this article easily.
Author Response
Reviewer #1
Dear authors,
The topic of this article is interesting. The information about "Biology and Molecular Targets of CBD" and "Role of CBD in Cancer treatment, pre-Clinical studies: CBD as a Component of Combination Therapies for Cancer" were presented followed by "Clinical Studies of CBD in Cancer" like the literature report. There was no critical evaluation of that information.
Therefore, the critical evaluation should be added to demonstrate the knowledge and understanding of the authors. The reader will get the advantage from this article easily.
Response:
We would like to thank the reviewer for highlighting the interest in the topic of CBD use in cancer and the benefits that readers will gain from our article “the reader will get the advantage from this article easily.” We appreciate the suggestion to add additional critical evaluation of the clinical trial results. To address this concern, we have added further discussion and evaluation to the “clinical studies of CBD in cancer” and "concluding remarks" sections.
Reviewer 2 Report
The present paper does not represent any relevant contribution to the field due to the high number of similar articles already published in the literature. To this concern, the manuscript can not be published in this journal.
Author Response
Reviewer #2
The present paper does not represent any relevant contribution to the field due to the high number of similar articles already published in the literature. To this concern, the manuscript cannot be published in this journal.
Response:
We would like to thank the reviewer for their efforts in providing an assessment of our manuscript and for pointing out that there is a high number of recent articles on this topic. We respectfully disagree with this reviewer’s view. Firstly, the existence of a high number of similar articles in the literature indicates that a high level of interest exists in this topic among researchers. Second, our manuscript offers three unique contributions to the field that are not found in similar articles.
- Our review bridges the gap between the molecular mechanism focused pre-clinical studies of CBD and the palliative care focused clinical trials of CBD whereas many previously published reviews only focus on one aspect or the other. This analysis highlights the potential for future clinical studies of the anti-cancer properties of CBD combination therapies, and we provide a critical appraisal of this topic that may be useful in the design of such studies.
- Our review includes novel meta-analysis of differential gene expression in CBD treated cancer cells from multiple cancer types (Fig 4). CBD has been described to have many various molecular effects on cancer cells as we have reviewed in the section pertaining to pre-clinical studies. Therefore, our analysis reporting the common molecular features of CBD treatment in three different cancer types may be useful in designing future mechanistic studies or selecting drugs for combination therapy with CBD.
- Furthermore, we have made a novel comparison of the molecular effects of CBD in cancer with the molecular effects of THC (Fig 5-6). We feel this is a relevant contribution because CBD and THC are frequently used together in the clinic and our analysis highlights the need for clinical studies of CBD without THC.
We have edited the manuscript to emphasize these contributions and we hope the reviewer will find our revised manuscript acceptable.
Reviewer 3 Report
In Green et al., the authors reviewed the therapeutic roles of cannabidiol (CBD) in cancers. This manuscript introduced CBD’s physiological pathway and molecular targets, reviewed a wide range of preclinical studies of various cancer models, listed almost all the clinical trials up-to-date, and discussed perspectives and limitations of using CBD as an effective anti-cancer drug candidate beyond a purely palliative or combination therapy.
Following a clear logic tread, the review manuscript provides a complete and relevant overview of CBD-related oncology therapies. The therapeutic potential of CBD is highlighted with a description of many relevant studies and its challenges are also sufficiently evaluated and discussed. It will elucidate basic concepts and knowledge for both newcomers and experienced scientists in this field. Overall, it meets the scope and standard of International Journal of Molecular Sciences, especially in the section on Molecular Oncology, and I recommend acceptance of this manuscript.
Author Response
In Green et al., the authors reviewed the therapeutic roles of cannabidiol (CBD) in cancers. This manuscript introduced CBD’s physiological pathway and molecular targets, reviewed a wide range of preclinical studies of various cancer models, listed almost all the clinical trials up-to-date, and discussed perspectives and limitations of using CBD as an effective anti-cancer drug candidate beyond a purely palliative or combination therapy.
Following a clear logic tread, the review manuscript provides a complete and relevant overview of CBD-related oncology therapies. The therapeutic potential of CBD is highlighted with a description of many relevant studies and its challenges are also sufficiently evaluated and discussed. It will elucidate basic concepts and knowledge for both newcomers and experienced scientists in this field. Overall, it meets the scope and standard of International Journal of Molecular Sciences, especially in the section on Molecular Oncology, and I recommend acceptance of this manuscript.
Response:
We thank the reviewer for their thorough evaluation of our manuscript. We appreciate the reviewer identifying the strengths of our manuscript including, “following a clear logic tread”, and “It will elucidate basic concepts and knowledge for both newcomers and experienced scientists in this field.”
Round 2
Reviewer 1 Report
Dear Authors,
This revision shows the improvement of the article. It could be accepted for publication after a minor spell check.
Author Response
We thank the reviewer for his evaluation of our manuscript. To address the reviewer’s concerns, we have performed a spelling and grammar check and corrected several spelling, capitalization, and typographical errors.
Reviewer 2 Report
Although the authors highlighted different interesting aspects in the letter, the manuscript still lacks a clear discussion on the importance of these points. A comparison with the recent literature published on similar aspects would have been appreciable, in order to underline the importance of the proposed manuscript. However, the methodology followed to perform the literature search is not clear, because it is not mentioned at all. The lack of a methodology description (inclusion criteria, databases evaluated, exclusion criteria etc.) contributes generating confusion. To this concern, the PRISMA workflow is an excellent tool to produce a review article. Finally, a consideration of the different treatments administered should be included, in order to provide a more complete overview of the current state of art.
Author Response
We would like to thank the reviewer for his thorough evaluation of our manuscript. We appreciate the reviewer’s suggestions to improve our manuscript and strengthen its impact. To address the reviewer’s concerns we have added a comparison between our manuscript and recent literature published on the topic of CBD use in cancer to the discussion section. Furthermore, we have added a methodology section describing our method for literature search and evaluation. Finally, we have added a discussion of the treatments currently used for symptom palliation in cancer patients to better define the unmet need and the potential of CBD. We hope the reviewer will find our revised manuscript acceptable for publication.
Round 3
Reviewer 2 Report
Since the authors satisfactorily addressed all the reviewer's concerns, the manuscript can be accepted for publication in this journal.